# Efficient Activation of Peroxymonosulfate by Biochar-Loaded Zero-Valent Copper for Enrofloxacin Degradation: Singlet Oxygen-Dominated Oxidation Process

**DOI:** 10.3390/nano12162842

**Published:** 2022-08-18

**Authors:** Jiang Zhao, Tianyin Chen, Cheng Hou, Baorong Huang, Jiawen Du, Nengqian Liu, Xuefei Zhou, Yalei Zhang

**Affiliations:** 1State Key Laboratory of Pollution Control and Resources Reuse, College of Environmental Science and Engineering, Tongji University, Shanghai 200092, China; 2Shanghai Institute of Pollution Control and Ecological Security, Shanghai 200092, China

**Keywords:** peroxymonosulfate, biochar-loaded zero-valent copper, enrofloxacin, singlet oxygen, superoxide radical

## Abstract

**Simple Summary:**

The misuse of antibiotics has caused ecological and human health risks on a global scale. Peroxymonosulfate can generate reactive oxygen species with extremely strong oxidative properties, which can degrade most types of antibiotics. For efficient removal of antibiotics in the aqueous environment, an economic biochar-loaded zero-valent copper was prepared by a simple pyrolysis method to activate peroxymonosulfate so that it can generate reactive oxygen species to oxidative and degrade the typical antibiotics, enrofloxacin. It was shown that complete degradation of enrofloxacin could be achieved within 30 min using biochar-loaded zero-valent copper to activate peroxymonosulfate, and the process of reactive oxygen species generation and the degradation pathway of enrofloxacin were also revealed.

**Abstract:**

The removal of contaminants of emerging concern (CECs) has become a hot research topic in the field of environmental engineering in recent years. In this work, a simple pyrolysis method was designed to prepare a high-performance biochar-loaded zero-valent copper (CuC) material for the catalytic degradation of antibiotics ENR by PMS. The results showed that 10 mg/L of ENR was completely removed within 30 min at an initial pH of 3, CuC 0.3 g/L, and PMS 2 mmol/L. Further studies confirmed that the reactive oxygen species (ROS) involved in ENR degradation are ·OH, ${SO}_{4}^{-}$·, ^1^O_2_, and $O_{2}^{-}$. Among them, ^1^O_2_ played a major role in degradation, whereas $O_{2}^{-}$· played a key role in the indirect generation of ^1^O_2_. On the one hand, CuC adsorbed and activated PMS to generate ·OH, ${\mathrm{SO}}_{4}^{-}$· and $O_{2}^{-}$·. $O_{2}^{-}$· was unstable and reacted rapidly with H_2_O and ·OH to generate large amounts of ^1^O_2_. On the other hand, both the self-decomposition of PMS and direct activation of PMS by C=O on biochar also generated ^1^O_2_. Five byproducts were generated during degradation and eventually mineralized to CO_2_, H_2_O, ${NO}_{3}^{-}$, and F^−^. This study provides a facile strategy and new insights into the biochar-loaded zero-valent transition-metal-catalyzed PMS degradation of CECs.

## 1. Introduction

In recent years, contaminants of emerging concern (CECs) such as drugs and personal care products (PPCPs), endocrine disruptors (EDCs), and perfluorinated compounds (PFCs) have become a research hotspot in the field of water treatment because they are difficult to remove completely by conventional water and wastewater treatment processes [1]. Antibiotics, which are widely used as bactericides in human and animal health care, are the most common class of PPCPs. They persist stably in nature after discharge and may pose a risk to the environment and public health, even at trace levels [2]. Enrofloxacin (ENR) is a synthetic antibiotic that belongs to the third generation of fluoroquinolones. It is frequently used in animal medicine owing to its broad-spectrum and powerful bactericidal effects [3]. ENR is also difficult to biodegrade and has a half-life of 3–9 years [4], leading to its easy entry into aquatic environments through poultry manure and feed, including livestock and poultry breeding wastewater [5], wastewater treatment plant effluent [6], surface water [7], groundwater [8], and tap water [9], thus becoming a major threat to human health and aquatic ecosystems [10].

In terms of antibiotic degradation in aquatic environments, advanced oxidation processes (AOPs), especially sulfate radical-based AOPs (SR-AOPs), have become important because of their high reaction rate, strong oxidation capacity, wide pH-applicable range, and long radical half-life (30–40 μs) [11,12]. Hard-to-degrade organic pollutants can be oxidized to low-toxic or non-toxic byproducts and even further mineralized to CO_2_ and H_2_O. Peroxydisulfate (PDS) and peroxymonosulfate (PMS) are the two most common persulfate (PS) to generate sulfate radicals (${SO}_{4}^{-}$·). Compared to the symmetrical structure of PDS, the asymmetrical structure of PMS leads to a partial positive charge on its peroxide bond, which is more susceptible to attack by nucleophilic reagents [13]. Therefore, the reactivity of PMS is higher than that of PDS. In addition to ${SO}_{4}^{-}$·, PMS can also generate other reactive oxygen species (ROS), including ·OH and $O_{2}^{-}$· in a radical pathway and ^1^O_2_ in a non-radical pathway. The key to the application of SR-AOPs is the activation of PS. Conventional activation methods include energy (e.g., thermal, ultraviolet, and microwave), transition metals and their oxides, and carbon materials (e.g., carbon nanotubes, graphene, and biochar) [14,15,16,17]. Based on the principle of the Fenton reaction, much research has been conducted on transition metal-activating PMS systems [18]. Most studies have focused on the activation of PMS by low-valent transition metal ions (e.g., Fe^2+^ [19], Co^2+^ [20], and Cu^+^ [21]) and transition metal oxides (e.g., Fe_2_O_3_ [22] and CuCo_2_O_4_ [23]), whereas zero-valent transition metal catalysts have rarely been investigated. Zero-valent transition metals are also promising activators of PMS [24]. Related studies have found that the use of zero-valent Fe to activate PMS for pollutant degradation was even more efficient than the use of Fe^2+^ [24,25,26]. The redox properties of Cu are similar to those of Fe, and Cu has been proven to be more stable than Fe [27,28]. It has been shown that zero-valent Cu (Cu^0^) can directly activate PMS to generate ${\mathrm{SO}}_{4}^{-}$· [29], and in jar experiments, zero-valent Cu even has higher efficiency in activating PMS than zero-valent iron because of zero-valent iron’s magnetic susceptibility [30]. Therefore, the use of Cu^0^ activating PMS system is worthy of researcher’s further study.

However, the use of Cu^0^/PMS systems has some drawbacks, such as the common use of nano-sized or micron-sized copper [30,31,32], which tends to agglomerate in water. The metal ions generated by the Cu^0^ activating PMS process can lead to secondary contamination of water [28]. To overcome these drawbacks, some researchers have used the concept of metal organic frameworks (MOFs) to prepare C@Cu-Ni [33] to reduce the leaching of metal ions. Moreover, some researchers have used polyurethane foam loaded with Cu^0^ [34], in which Cu is uniformly dispersed on the surface of polyurethane foam to reduce its agglomeration effect. However, the high preparation costs of MOFs, strict requirements for synthesis conditions, and poor reproducibility of the prepared products limit their practical application [35,36]. The pollution generated during the production and use of conventional polyurethane materials can threaten the environment and human health [37]. In recent years, biochar (BC)-based transition metal composites have received increasing attention and are regarded as excellent materials that can effectively reduce metal agglomeration and lower metal ion leaching rates. Additionally, transition metal-based BC composites possess the characteristics of a simple preparation operation, a wide range of sources of raw materials, and stable PMS activation performance [38,39]. Nevertheless, few studies have reported the activation of PMS by BC-loaded Cu^0^ (CuC), and the mechanism by which the CuC/PMS system degrades pollutants still needs further investigation.

The goal of this study was to provide a simple pyrolysis method to directly prepare CuC to effectively activate PMS for the rapid removal of ENR. The CuC was characterized using X-ray diffraction (XRD), X-ray photoelectron spectroscopy (XPS), scanning electron microscopy (SEM), and energy dispersive spectroscopy (EDS). The degradation performance of the systems was investigated under different catalyst dosages, PMS dosages, pH values, and the presence of common coexisting ions and organic matter in the aqueous environment. The potential mechanism of CuC/PMS degradation of ENR was systematically investigated through quenching experiments, electron spin resonance (EPR), and PMS decomposition experiments in the presence of quenching agents, and possible pollutant degradation products were inferred by using liquid chromatography–mass spectrometry (LC-MS) analysis.

## 2. Materials and Methods

### 2.1. Chemicals

Copper chloride dihydrate (CuCl_2_·2H_2_O), methanol (99.5%, MeOH), and ENR (100 mg) were purchased from Shanghai Maclean Biochemical Co., Ltd. (Shanghai, China). Tert-butanol (≥99%, TBA), p-benzoquinone (99%, PBQ), peroxymonosulfate (≥42%, KHSO_5_·0.5KHSO_4_·0.5K_2_SO_4_), potassium iodide (>99.0%, KI), sodium thiosulfate (99.99%), NaOH, H_2_SO_4_, NaHCO_3_, and Na_2_SO_4_ were purchased from Shanghai Aladdin Bio-Chem Technology Co., Ltd. (Shanghai, China). L-histidine (≥99%, L-His) was purchased from Sigma Aldrich (Shanghai, China) Trading Co. The straw was obtained from Jiangsu Lianfeng Agricultural Products Deep Processing Co. (Zhangjiagang, China). The above reagents and other reagents used in this study were of analytical grade or higher and were not purified for direct use.

### 2.2. Preparation of CuC

Deionized water (40 mL), 0.4262 g copper chloride dihydrate, and 1 g of maize straw powder were added to a 250 mL beaker. After adding a magnetic rotor, the beaker was placed in a constant-temperature magnetic stirrer at 400 rpm for 24 h. The beaker was removed and sealed with tin foil, 4–6 holes were made in the tin foil for ventilation, and the beaker was placed in an oven at 80 °C for 24 h. The dried material was transferred to a crucible and fired in a tube furnace at 700 °C under a nitrogen atmosphere. The prepared CuC was ground in a mortar and sieved through a 200-mesh sieve. After sieving, samples were placed in vacuum-sealed bags.

### 2.3. Jar Experiments

A series of batch experiments were conducted in 250-mL beakers. First, 100 mL of deionized water and a magnetic rotor were added to a 250-mL beaker, and the beaker was placed in a hexagonal thermostatic magnetic stirrer (900 rpm, 25 ± 1 °C). After the temperature was stable, ENR (10 mg/L) and PMS (2 mmol/L) were added to the deionized water, and the pH was 3.12 ± 0.1. For the quenching experiments, the scavengers were also added at this time. The activation of PMS was triggered by the addition of CuC (0.2 g/L). Water samples were then taken sequentially at intervals of 1, 2, 5, 10, 30, and 60 min and filtered through a 0.45-μm membrane. Na_2_S_3_O_4_ (0.2 mol/L) was added to the samples to prevent further reactions prior to the analysis. The effects of CuC dosage (0.05–0.4 g/L), PMS dosage (0.5–4 mmol/L), initial pH (2–11), and concentrations of ${HCO}_{3}^{-}$ (0–150 mmol/L), ${SO}_{4}^{2-}$ (0–5 mmol/L), HA (0–0 mmol/L), and ${Cl}^{-}$ (0–150 mmol/L) on the removal efficiency were investigated. The pH of the solutions was adjusted using H_2_SO_4_ and NaOH solutions. All experiments were conducted two or more times.

### 2.4. Determination of PMS Concentration

The PMS concentration in the solution was determined by the yellow iodine color formed from KI. NaHCO_3_ (1.5 g) and KI (30 g) were added to 300 mL of deionized water and mixed thoroughly as a background solution. A portion of the reaction solution (0.5 mL) was extracted, and then added in a 10-mL colorimetric tube. The reaction solution was fixed to a scale using a background solution and shaken for 15 min. The final analysis was performed at a wavelength of 352 nm by using a UV spectrophotometer [40].

### 2.5. Characterization Methods

The XRD patterns of the samples were analyzed using a D8-Advance X-ray diffractometer (Bruker). The morphological structures and chemical compositions of the samples were observed using SEM (Hitachi Regulus 8100, Tokyo, Japan). The composition and chemical state of the elements were determined by XPS (Thermo Scientific K-Alpha, Waltham, MA, USA). The surface charge of the catalysts was determined by measuring the ζ potential in Milli-Q ultrapure water using a Zetasizer (Nano ZS90, Malvern, UK). The Cu content in CuC was detected using an inductively coupled plasma optical emission spectrometer (ICP-OES, Perkin Elmer Optima 8000, Waltham, MA, USA).

### 2.6. Analytical Methods

The ENR concentration was measured using an Agilent 1200 Infinity high-performance liquid chromatography (HPLC) instrument (Agilent, Santa Clara, CA, USA) equipped with a UV detector and an Agilent ZORBAX SB-C18 column (4.6 mm × 250 mm, 5 μm) (detailed information provided in Appendix A). The transformation products of ENR were measured by ultra-high-performance liquid chromatography (UPLC, Agilent 1290) and tandem mass spectrometry (Q-Exactive Plus MS/MS, Agilent QTOF 6550, USA). Electron paramagnetic resonance (EPR, Bruker EMX Plus, Rheinstetten, Germany) analysis was performed to detect the generated radicals.

## 3. Result and Discussions

### 3.1. Characterization of the Samples

Figure 1a shows the XRD patterns of the BC and the prepared CuC material. The diffraction peak of Cu^0^ is clearly observed in the XRD spectrum of CuC (PDF # 04-0836), indicating that it is mainly Cu^0^ loaded onto the BC. The morphology of the prepared CuC was characterized using SEM and the results are shown in Figure 1. BC with a tubular structure is clearly observed in Figure 1c. Combined with the EDS distribution layer diagram, it can be confirmed that the stratiform surface of the catalytic material are loaded with Cu particles. The Cu particles are uniformly distributed on the surface of the biochar, indicating that the Cu and biochar underwent good bonding during the preparation process (Figure 1b,d). In addition, the elemental test images (Figure 1e) revealed that the main elements in CuC are C, O, Cu, K, and Cl. The presence of KCl in the XRD spectra of both BC and CuC (PDF # 41-1476) suggests that K and Cl may have originated from the KCl fertilizer used in the straw fertilization process (Figure 1a). In combination with ICP-OES testing, the theoretical mass fraction of copper in the catalyst was determined to be 24.6%.

### 3.2. Degradation of ENR in Different Systems

The degradation effects of ENR in different reaction systems are shown in Figure 2. According to previous studies, BC is usually used as an adsorbent for organic matter because of its special pore structure [41]. However, the degradation efficiency of ENR in the presence of BC and CuC alone are only 4.6% and 8.7%, respectively, within 60 min (Figure 2). This indicates that the percentage of ENR removed by the adsorption of BC is extremely low in the system, and the loading of Cu ions (such as Cu^+^ and Cu^2+^) can form a complexation reaction between transition metal ions and organic matter, which increases the chemisorption of ENR [42]. PMS are considered as a strong oxidant, but it exhibits low efficiency when it reacts directly with organic pollutants [15]. PMS can be activated to ROS by UV activation [43], thermal activation [44], and other pathways. Therefore, the degradation efficiency of ENR can reach 24.6% when PMS is used alone. BC is often used as an effective catalyst for PMS-based AOPs [16], and it can directly activate PMS through C=O bonding to produce ^1^O_2_, which leads to the oxidative decomposition of organic pollutants [29]. However, the degradation efficiency of ENR in the BC/PMS system are close to that of PMS alone up to 30 min, and by 60 min. That are less than 10% higher than that of PMS alone. This implies that the catalytic activation of PMS by C=O is limited in terms of the rapid degradation of the organic pollutants. The degradation efficiency of ENR in the (Cu + BC)/PMS system (using copper powder) reached 89%, indicating that Cu^0^ is the main species for catalytic activation of PMS in this system. However, the (Cu + BC)/PMS system could not rapidly and completely degrade the pollutants within 30 min because of the competitive relationship rather than synergetic relationship between Cu and BC for the adsorbent. There are problems with physically mixing Cu^0^ and BC, such as difficult Cu recovery and secondary contamination in the water body by Cu ions after the reaction [45]. In contrast, the CuC/PMS system has a higher degradation efficiency and can completely degrade ENR within 30 min. It also has a lower leaching rate and a good recycling rate [29]. The above experiments illustrate that Cu loading on BC made from straw to activate PMS can remove organic pollution from wastewater quickly and effectively.

### 3.3. Degradation of ENR by the CuC/PMS System under Various Experimental Conditions

#### 3.3.1. CuC and PMS Dosage

The effect of the amount of CuC on the ENR degradation efficiency was investigated when the PMS dosage was fixed. As shown in Figure 3a, the amount of CuC in the range of 0.05–0.4 g/L had a good degradation effect on ENR. Upon increasing the CuC dosage from 0.05 g/L to 0.2 g/L, the degradation efficiency of ENR increased from 87% to 100% within 60 min. The active sites on the surface of CuC (including Cu [29], oxygen-containing functional groups on the biochar surface [39], defective sites [46] and persistent radicals [47]) can catalyze PMS to generate ROS and then oxidize ENR. Therefore, with increasing CuC, the concentration of ROS involved in oxidation also increased, which improved the degradation efficiency of ENR. When the amount of CuC increased from 0.3 g/L to 0.4 g/L, the degradation rate of ENR is not improved significantly, which is attributed to the saturation of active sites on CuC for PMS.

The effect of the amount of PMS on the ENR degradation efficiency was examined when the CuC dosage was fixed. As can be seen from Figure 3b, increasing the PMS dosage from 0.5 mmol/L to 2 mmol/L significantly increased the degradation efficiency of ENR from 63% to 100% within 60 min. However, the degradation efficiency of ENR is similar within 60 min when the concentration increased from 2 mmol/L to 4 mmol/L. In the first 30 min, the degradation effect at 3 mmol/L is slightly better than that at 4 mmol/L. Raising the PMS dosage within a certain range could lead to an increase in ROS concentration, which would promote the degradation of ENR. However, excessive PMS transiently generates certain ROS (${\mathrm{SO}}_{4}^{-}$· and ·OH) with high concentrations, resulting in the self-quenching reactions shown in Equations (1) and (2) [48,49] to form ${SO}_{5}^{-}$· and ${HSO}_{4}^{-}$, respectively, with lower reactivities. The oxidation potential of ${SO}_{5}^{-}$· [E^0^ (${SO}_{5}^{-}$/${HSO}_{5}^{-}$) = 0.81 V] is lower than that of ${SO}_{4}^{-}$· [E^0^ (${SO}_{4}^{-}$/${HSO}_{5}^{-}$) = 2.6–3.1 V] [50], which affects the oxidative degradation efficiency of ENR.
(1)$$\mathrm{Cu}+{\;\mathrm{HSO}}_{5}^{-}\;\to {\;\mathrm{Cu}}^{+}+{\;\mathrm{SO}}_{4}^{-}\cdot$$
(2)$${\mathrm{SO}}_{4}^{-}\cdot +{\;\mathrm{OH}}^{-}\;\to \;\cdot \mathrm{OH}+{\mathrm{SO}}_{4}^{2-}$$
(3)$${\mathrm{HSO}}_{5}^{-}+\cdot \mathrm{OH}\;\to \;H_{2}O+{\;\mathrm{SO}}_{5}^{-}\cdot$$
(4)$${\mathrm{HSO}}_{5}^{-}+{\;\mathrm{SO}}_{4}^{-}\cdot \to {\;\mathrm{HSO}}_{4}^{-}+{\;\mathrm{SO}}_{5}^{-}\cdot$$

#### 3.3.2. pH

Different pH values not only affect the activation of PMS but also the subsequent reaction between ROS and pollutants [51]. Figure 4 shows that the CuC/PMS system efficiently degraded ENR in the pH range of 2 to 11 within 60 min. This also reflects a possible non-free radical pathway to degrade ENR in the system because the process is somewhat less affected by pH [52]. Generally, in the pH range of 2–9, the degradation efficiency increases with increasing pH in the PMS degradation system. Under strongly acidic conditions, H^+^ combines with the O–O bond in HSO_5_^−^ to form a strong hydrogen bond, inhibiting the activation of PMS [53]. However, owing to the different properties of catalysts and pollutants, the degradation effect at pH 3 is better than that at pH 5 in the system. Zeta potential tests of CuC were performed to investigate the cause. The results showed that when 2 < pH < 9, the surface of CuC is positively charged, which diminished with increasing pH (Appendix A). This indicates that at pH = 3, the positive effect of the positive electrical attraction of the CuC surface to PMS is probably stronger than the inhibitory effect of H^+^ binding to the O–O bond. The negative effect of the weakening of the positively charged CuC surface at pH = 5 is probably stronger than the positive effect of the decrease in H^+^. In other words, the electrical attraction of CuC to PMS and the strong hydrogen bonds formed by PMS under the action of H^+^ both affected the degradation efficiency of the system. Additionally, the degradation effect of ENR decreased when the pH increased from 9 to 11. Because the pK_a_ of PMS is 9.4, hydrolysis reactions occur in strongly alkaline environments [Equation (3)] [54]. This affects the efficiency of PMS in ROS production. Nevertheless, the CuC/PMS system exhibits an excellent degradation effect under acidic, neutral, and alkaline conditions. It also operates under a wide range of pH values.
(5)$${\mathrm{HSO}}_{5}^{-}+2{\mathrm{OH}}^{-}\to {\;H}_{2}O+2{\mathrm{SO}}_{4}^{2-}+{\;O}_{2}$$

#### 3.3.3. Coexisting Ions and Humic Acid

${SO}_{4}^{2-}$, ${HCO}_{3}^{-}$,${Cl}^{-}$, and humic acid are common coexisting ions and organic acids in water [55,56]. Under a series of concentration gradients, humic acid, ${SO}_{4}^{2-}$, and ${HCO}_{3}^{-}$ will reduce the degradation efficiency of ENR by CuC/PMS to a certain extent (Figure 5). This is because these components may adhere to the surface of the carbon-based catalyst in the reaction solution [29], affecting the adsorption of CuC and PMS and reducing the ROS generation efficiency. Humic acid and ${SO}_{4}^{2-}$ have a minimal inhibiting effect on the degradation rate, whereas ${HCO}_{3}^{-}$ has a somewhat stronger inhibiting effect the degradation rate. This may be because ${HCO}_{3}^{-}$ scavenges ·OH and ${SO}_{4}^{-}$· to form ·HCO_3_ [45,57], thus inhibiting the degradation efficiency of ENR. ${Cl}^{-}$ can react with ${HSO}_{5}^{-}$ to form HClO, which has a strong oxidizing power that can promote the degradation of ENR [58]. Therefore, the ENR degradation efficiency is slightly influenced by the presence of coexisting ions and organic acids.

### 3.4. Mechanism Study

#### 3.4.1. Identification of the Reactive Oxygen Species

The ROS involved in the degradation of ENR in the CuC/PMS system were identified by EPR and quenching tests. Degradation reactions were quenched using excess amounts of scavengers, including tert-butanol (TBA, 4 mmol/L), methanol (MeOH, 4 mmol/L), L-histidine (LHD, 10 mmol/L), and p-benzoquinone (PBQ, 5 mmol/L) for ·OH (k = 3.8–7.6 × 10^8^ M^−1^·s^−1^), ${SO}_{4}^{-}$· (k = 2.5 × 10^7^ M^−1^·s^−1^), and ^1^O_2_ [59] and $O_{2}^{-}$· (k = 0.9–1.0 × 10^9^ M^−1^·s^−1^), respectively. The effect of the pH on the degradation reaction was systematically investigated. As shown in Figure 6a–c, when scavenging ·OH and ${SO}_{4}^{-}$·, the removal rates of ENR in the MeOH and TBA quenching systems are (43.59%, 45.1%), (67.92%, 63.72%), and (68.11%, 70.18%) at pH = 3, 7, and 9, respectively. Because MeOH can effectively scavenge ·OH and ${SO}_{4}^{-}$·, whereas TBA mainly scavenges ·OH [60], the similar removal rates when scavenging separately indicate that ·OH is more dominant in the degradation process than ${SO}_{4}^{-}$·, which is probably owing to the rapid conversion of ${SO}_{4}^{-}$· to ·OH (k = 6.5 × 10^7^ M^−1^·s^−1^) [61] The conversion allowed more ·OH in the system to participate in ENR degradation. However, ENR degradation in the reaction systems at pH = 3, 7, and 9 are severely inhibited when scavenging ^1^O_2_, and the degradation rates are only 7.02%, 5.46%, and 8.43%, respectively. The ENR degradation also decreased substantially when quenching $O_{2}^{-}$·, with degradation rates of 15.98%, 19.41%, and 27.77%, respectively, revealing that ^1^O_2_ and $O_{2}^{-}$· may be the main ROS for ENR degradation in the CuC/PMS system. Meanwhile, the scavenging of ^1^O_2_ and $O_{2}^{-}$· both had a strong inhibitory effect on ENR degradation. The scavenging of ^1^O_2_ had a stronger inhibitory effect on ENR degradation than $O_{2}^{-}$·, probably because $O_{2}^{-}$· is the main precursor for the generation of ^1^O_2_ [Equation (10)] [62]. Additionally, the concentration of PMS in the reaction was measured. In addition, the PMS concentration of the reaction was measured. The results showed that in the LHD and PBQ scavenging systems, the PMS concentration decreased rapidly by 81.67% and 94.54%, respectively, within 10 min (Figure 6d). This phenomenon ruled out the possibility that the decreased degradation rate of ENR is owing to the hydrophobicity of organic scavengers blocking the contact between PMS and CuC [63], and it proved the previously mentioned reliability of the quenching tests.
(6)O2−·+2H2O → H2O2+OH−+O 12

EPR analysis directly revealed the ROS generated in the CuC/PMS system. 5,5-Dimethyl-pyrroline-N-oxide (DMPO) and 2,2,6,6-tetramethyle-4-piperidone (TEMP) were used as spin-trapping agents in the CuC/PMS systems (Figure 6e,f). Initially, the typical four-line peak of DMPO-·OH in the CuC/PMS system in the form of 1:2:2:1 are clearly visible. The intensity of the characteristic peak of the DMPO- ${\mathrm{SO}}_{4}^{-}$· adduct (in the form of 1:1:1:1:1:1:1) is much weaker than that of DMPO-·OH [64], which further indicated that most of the ${SO}_{4}^{-}$· in the CuC/PMS system are converted to ·OH. Secondly, the typical 1:1:1 triplet signal peak of TEMP-^1^O_2_ [65] and the characteristic peak of $O_{2}^{-}$· were observed in the CuC/PMS system [66]. However, in the PMS system, only a weak TEMP-^1^O_2_ peak were observed and no spectrum of $O_{2}^{-}$· were found. This phenomenon can be reasonably explained by the fact that PMS can self-decompose to produce ^1^O_2_ [67], whereas Cu in CuC can effectively catalyze PMS to generate $O_{2}^{-}$·, and the rapid reaction of $O_{2}^{-}$· generates a large amount of ^1^O_2_ [65], resulting in a substantial increase in the degradation efficiency of ENR. This inference is supported by the 24.6% degradation effect of the PMS system alone as compared to the complete degradation of ENR by the CuC/PMS system within 30 min (Figure 3). Combining the quenching tests and EPR spectra analysis, it is proposed that in the CuC/PMS system, ·OH, ${\mathrm{SO}}_{4}^{-}$·, ^1^O_2_, and $O_{2}^{-}$· are all involved in the degradation process. Among all the ROS, ^1^O_2_ is the main ROS for degrading pollutants in the pollutant degradation process, whereas $O_{2}^{-}$· plays a key role as the main precursor substance in bridging the ^1^O_2_ reaction.

#### 3.4.2. Activation Process

In view of the important role of CuC in activating PMS in the CuC/PMS system, XPS analysis was used to characterize CuC before and after the degradation reaction (Figure 7) to further elucidate the mechanism of PMS activation. The pre-reaction the Cu 2p spectrum could be mainly deconvoluted into four peaks (Figure 7a), corresponding to Cu 2p_1/2_ (952.66 eV) and Cu 2p_3/2_ (932.74 eV) for Cu^0^ (55.8%) and to Cu 2p_1/2_ (950.15 eV) and Cu 2p_3/2_ (930.33 eV) for Cu/Cu^+^ (25.6%). In general, the major peaks in the Cu/Cu^+^ region are indistinguishable because the binding energy values are very similar. Related studies suggest that Cu^+^ are probably derived from the oxidation of Cu^0^ from the material surface, and thus its presence can be considered negligible [68]. The post-reaction Cu 2p spectrum was decomposed into two peaks at 932.43 and 952.22 eV (Figure 7b), which could be assigned to the presence of Cu^0^. Combined with the EDS results, the relative amount of Cu in CuC before the reaction is 4.8 times higher than that after the reaction, indicating the depletion of Cu in CuC (Table 1). Specifically, Cu^0^ provides electrons to convert ${HSO}_{5}^{-}$ to ${SO}_{4}^{-}$·, with the simultaneous generation of Cu^+^. Cu^+^ reacts with HSO_5_^−^ to form ${SO}_{4}^{-}$· and Cu^2+^. Cu^2+^ then converts ${HSO}_{5}^{-}$ to $O_{2}^{-}$· [Equations (4) and (5)]. The deconvolution of C 1s before and after the CuC reaction revealed that 9.5% of C=O is almost completely consumed after the reaction (Figure 7c,d). This may be attributed to the involvement of C=O in the direct activation of PMS to generate ${SO}_{4}^{-}$· and ^1^O_2_ [16,29].

The above phenomena indicate that there are three possible sources of ^1^O_2_ in the CuC/PMS system: PMS self-decomposition (Equation (10)), PMS directly activated by C=O, and PMS indirectly activated by Cu^2+^ (Equations (6) and (8)). Combined with the different degradation effects of PMS and the BC/PMS and CuC/PMS systems in the kinetic degradation reaction, this suggests that the predominant pathway in this study is the indirect generation of ^1^O_2_ by Cu^2+^-activating PMS. ^1^O_2_ is an ROS that is highly adaptable to different water bodies [69] and is less influenced by pH [70]. Therefore, CuC/PMS had an efficient degradation effect over a wide pH range and under the influence of different coexisting ions.

Based on the above analysis, the main mechanism of CuC activation of PMS includes the following: (1) Cu^0^ provides electrons to convert ${HSO}_{5}^{-}$ to ${SO}_{4}^{-}$·, and most of the ${SO}_{4}^{-}$· is converted into ·OH. (2) Cu^+^ reacts with ${HSO}_{5}^{-}$ to produce ${SO}_{4}^{-}$· and Cu^2+^. (3) Cu^2+^ reacts with ${HSO}_{5}^{-}$ to produce $O_{2}^{-}$· and Cu^+^· (4) $O_{2}^{-}$· is unstable and can react rapidly with H_2_O and ·OH to produce large amounts of ^1^O_2_. PMS self-decomposition and direct activation by C=O can also produce some ^1^O_2_. 

For clarity, the mechanism of degradation of ENR by the CuC/PMS system is summarized in Figure 8.
(7)$${\mathrm{Cu}}^{+}+{\;\mathrm{HSO}}_{5}^{-}\;\to {\;\mathrm{Cu}}^{2+}+{\;\mathrm{SO}}_{4}^{-}\cdot +{\;\mathrm{OH}}^{-}$$
(8)$${\mathrm{Cu}}^{2+}+{\;\mathrm{HSO}}_{5}^{-}+{\;H}_{2}O\to {\;O}_{2}^{-}\cdot +{\;\mathrm{Cu}}^{+}+{\;\mathrm{SO}}_{4}^{2-}+3H^{+}$$
(9)O2−·+·OH →O 12+ OH−
(10)HSO5−+ SO52− → HSO4−+ SO4−·+O 12
(11)O 12,·OH, SO4−,·O2−·+ENR → CO2+ H2O+ F−+NO3−

### 3.5. Proposed ENR Degradation Mechanism in CuC/PMS System

To better understand ENR degradation by CuC/PMS, LC-MS was uesd to identify the intermediate products during the process. According to Appendix A, comparing typical chromatograms at 0 min and 30 min during the reaction time, there is a significant ion peak, with *m*/*z* = 360 at the retention time *t* = 9.283 min before the reaction started, which is inferred to be ENR based on its molecular weight. After 30 min of reaction, the peak decreased significantly and split into two peaks, indicating that ENR was effectively degraded and transformed. After 30 min of reaction, five ENR intermediates were detected at retention times of 8.928, 9.536, 13.485, 15.916, and 18.143 min, with *m*/*z* values of 344, 362, 262, 334, and 390, respectively (Appendix A). The related secondary mass spectra are shown in the Appendix A. These results further demonstrate ENR degradation and reveal the possible degradation pathways of ENR.

As shown in Figure 9, pathway I involves the oxidation and loss of the piperazine ring [71]. The piperazine ring is opened and deaminated by the oxidation of the ROS to produce a ketone (BP1). The ketone groups are further oxidized to generate carboxyl groups, which are then decarboxylated (BP2). BP2 then undergoes a series of oxidation reactions to form BP3, leading to complete destruction of the piperazine group of ENR. Pathway II is mainly the ring opening of the quinolone moiety. Under the strong oxidation of ROS, the quinolone ring is progressively disrupted to produce BP4 and BP5 [72]. Eventually, BP5 and BP3 are further mineralized to CO_2_, H_2_O, ${NO}_{3}^{-}$, and F^−^ by a series of oxidation reactions in the presence of ROS. However, how to control the ecological toxicity of the degradation process of N-containing pollutants still needs more attention and in-depth research [73].

## 4. Conclusions

In this work, a simple pyrolysis method was designed to prepare a high-performance CuC material. Researchers investigated the effectiveness and mechanism of CuC activation of PMS to degrade ENR. Degradation experiment results of ENR in different systems indicate that Cu loading on BC made from straw to activate PMS can remove organic pollution from wastewater quickly and effectively. Under the conditions of initial pH = 3, CuC 0.3 g/L, and PMS 2 mmol/L, 10 mg/L of ENR was completely removed within 30 min. The CuC/PMS system exhibited good degradation in the range of pH = 2–11. The effects of the CuC dosage, PMS dosage, initial pH, and coexisting substances in water on the removal efficiency were evaluated. With the increase of CuC dosage or PMS dosage, the removal efficiency of ENR increased. The CuC/PMS system exhibits an excellent degradation effect with a wide range of initial pH.

The systemic EPR and quenching experiment were designed to investigate degradation mechanism of ENR in the CuC/PMS system. The results show that reactive oxygen species (ROS) involved in ENR degradation are ·OH, ${SO}_{4}^{-}$·, ^1^O_2_, and $O_{2}^{-}$·. Among them, ^1^O_2_ is the dominated ROS for degrading pollutants, which is generated mainly through the conversion of $O_{2}^{-}$·. On the one hand, CuC adsorbed and activated PMS to generate ·OH, ${SO}_{4}^{-}$· and $O_{2}^{-}$·. $O_{2}^{-}$· are unstable and reacted rapidly with H_2_O and ·OH to generate large amounts of ^1^O_2_. On the other hand, both the self-decomposition of PMS and direct activation of PMS by C=O on biochar also generated ^1^O_2_.

To better understand ENR degradation by CuC/PMS, LC-MS was used to identify the intermediate products. There are two pathways for the degradation of ENR in the CuC/PMS system: oxidation of the piperazine group and ring opening of the quinolone group. Five byproducts were generated during degradation and eventually mineralized to CO_2_, H_2_O, ${NO}_{3}^{-}$, and F^−^. This study provides a facile strategy and new insights into the biochar-loaded zero-valent transition metal-catalyzed PMS degradation of CECs.

## Figures and Tables

**Figure 1 nanomaterials-12-02842-f001:**
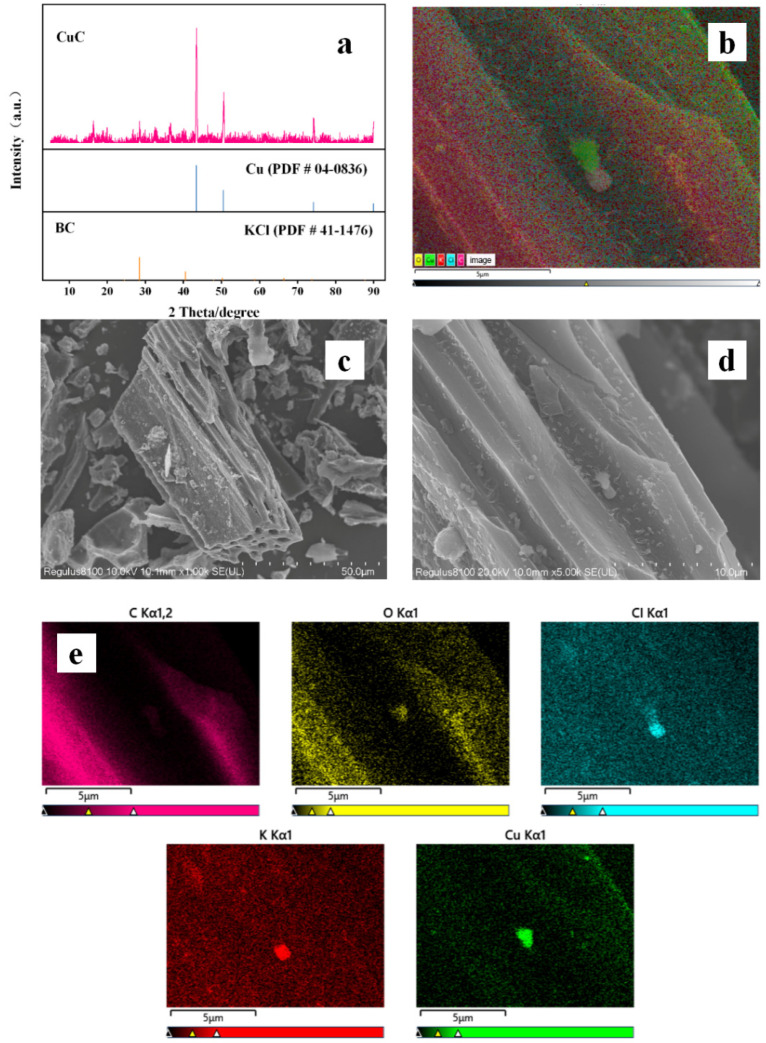
(**a**) XRD pattern of fresh CuC and BC, (**b**) distribution layer diagram of fresh CuC by EDS, (**c**,**d**) SEM images of fresh CuC, and (**e**) element mapping images of fresh CuC.

**Figure 2 nanomaterials-12-02842-f002:**
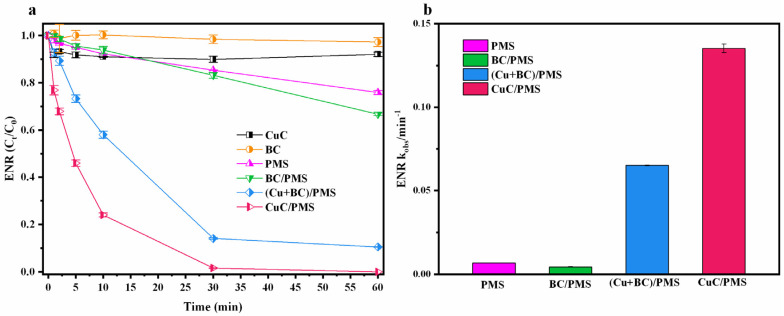
(**a**) The degradation of ENR in different systems, (**b**) k_obs_ in different systems. Experimental conditions: [catalyst]_0_ = 0.2 g/L ([Cu + BC]_0_ = 0.05 g/L Cu + 0.15 g/L BC), [ENR]_0_ = 10 mg/L, [PMS]_0_ = 2 mmol/L, initial pH = 3.10 (no pH adjustment), T = 25 °C.

**Figure 3 nanomaterials-12-02842-f003:**
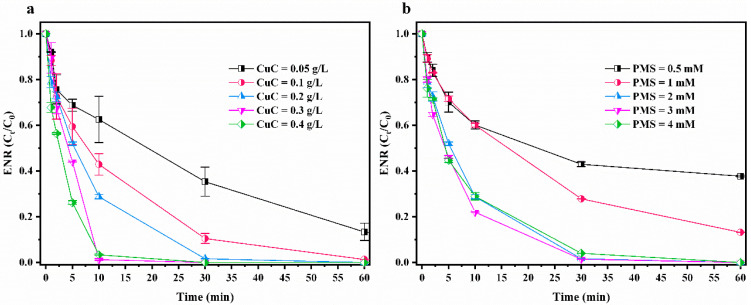
Effects of CuC and PMS dosage on ENR degradation by CuC/PMS. (**a**) degradation of ENR within 60 min using different CuC dosages and (**b**) degradation of ENR within 60 min by different PMS dosages. Experimental conditions: [CuC]_0_ = 0.2 g/L, [ENR]_0_ = 10 mg/L, [PMS]_0_ = 2 mmol/L, initial pH = 3.10 (no pH adjustment), T = 25 °C.

**Figure 4 nanomaterials-12-02842-f004:**
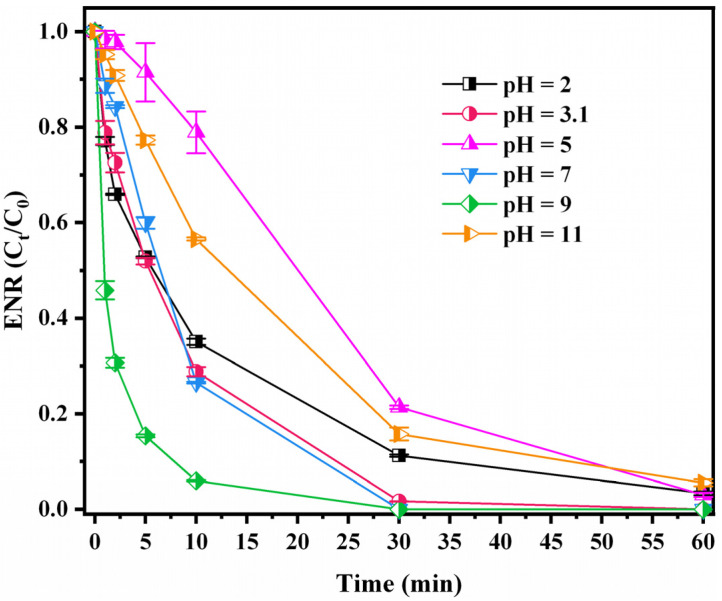
Effects of pH on ENR degradation by CuC/PMS. Experimental conditions: [CuC]_0_ = 0.2 g/L, [ENR]_0_ = 10 mg/L, [PMS]_0_ = 2 mmol/L, initial pH = 3.10 (no pH adjustment), T = 25 °C.

**Figure 5 nanomaterials-12-02842-f005:**
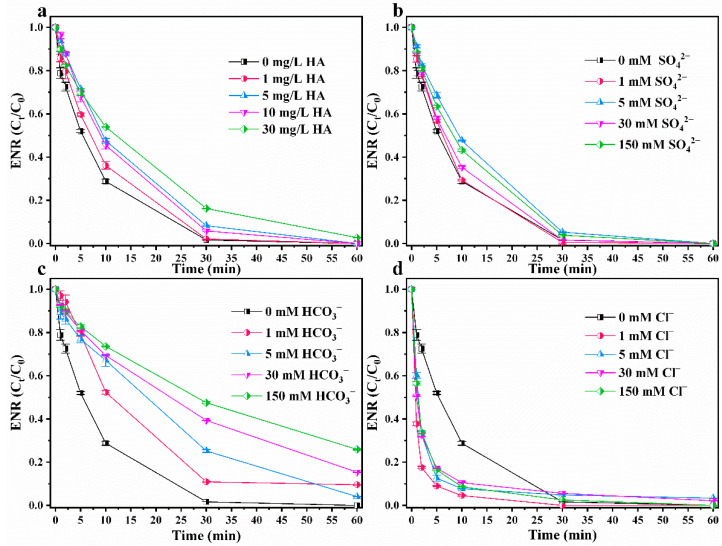
Effects of common inorganic anions and humic acids in water on ENR degradation by CuC/PMS. (**a**) humic acid, (**b**) ${SO}_{4}^{2-}$, (**c**) ${HCO}_{3}^{-}$, and (**d**) Cl^−^ Experimental conditions: [CuC]_0_ = 0.2 g/L, [ENR]_0_ = 10 mg/L, [PMS]_0_ = 2 mmol/L, initial pH = 3.10 (no pH adjustment), T = 25 °C.

**Figure 6 nanomaterials-12-02842-f006:**
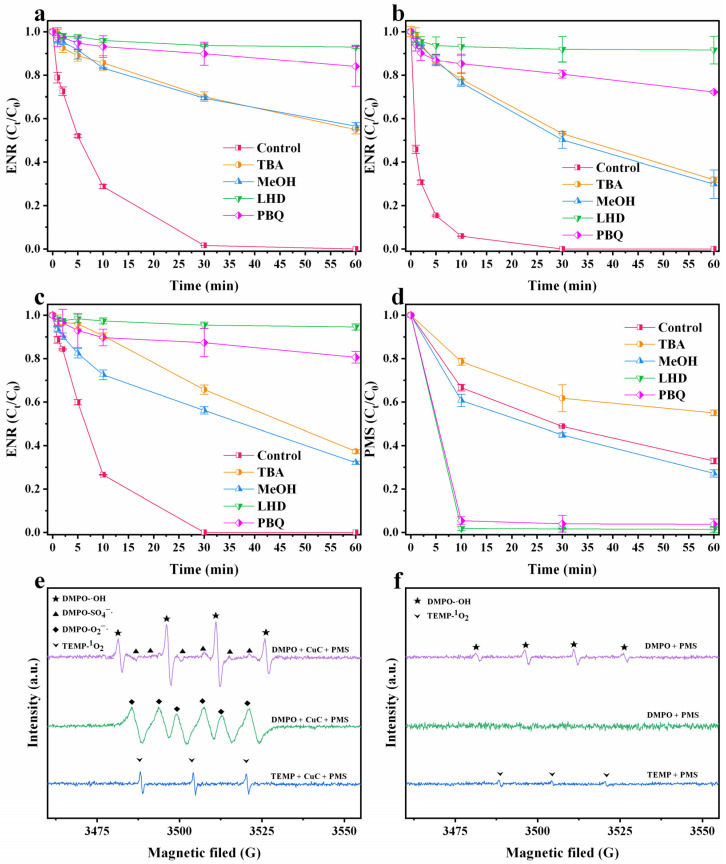
Effect of ROS scavengers (4 mmol/L TBA, 4 mmol/L MeOH, 10 mmol/L LHD and 5 mmol/L PBQ) on the removal of ENR by CuC/PMS at (**a**) pH = 3, (**b**) pH = 7, and (**c**) pH = 9. Experimental conditions: [CuC]_0_ = 0.2 g/L, [ENR]_0_ = 10 mg/L, [PMS]_0_ = 2 mmol/L, T = 25 °C. (**d**) Decomposition of PMS upon the removal of ENR by CuC/PMS. Experimental conditions: [CuC]_0_ = 0.2 g/L, [ENR]_0_ = 10 mg/L, [PMS]_0_ = 2 mmol/L, initial pH = 3.10, T = 25 °C. EPR spectra of (**e**) CuC/PMS and (**f**) PMS with the addition of DMPO and TEMP. Experimental conditions: [CuC]_0_ = 0.2 g/L, [ENR]_0_ = 10 mg/L, [PMS]_0_ = 2 mmol/L, initial pH = 3.10, T = 25 °C.

**Figure 7 nanomaterials-12-02842-f007:**
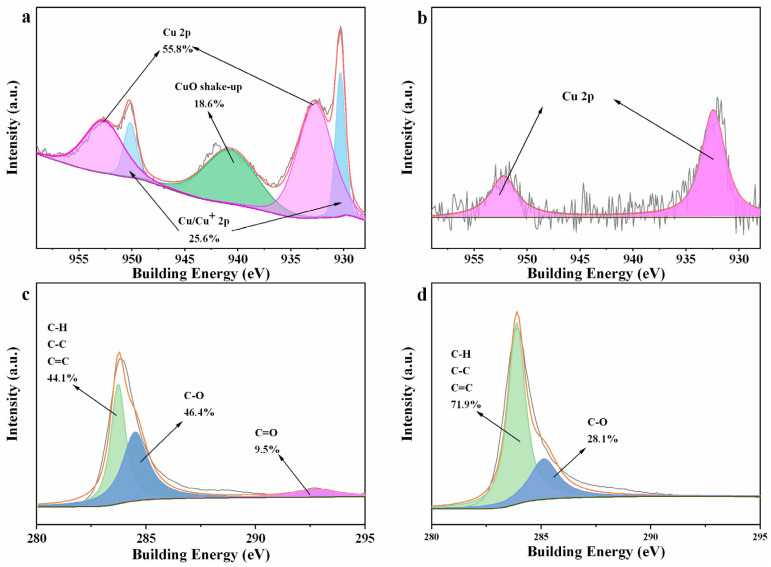
XPS spectra of (**a**) Cu 2p before reaction, (**b**) Cu 2p after reaction, (**c**) C 1s before reaction, and (**d**) C 1s after reaction.

**Figure 8 nanomaterials-12-02842-f008:**
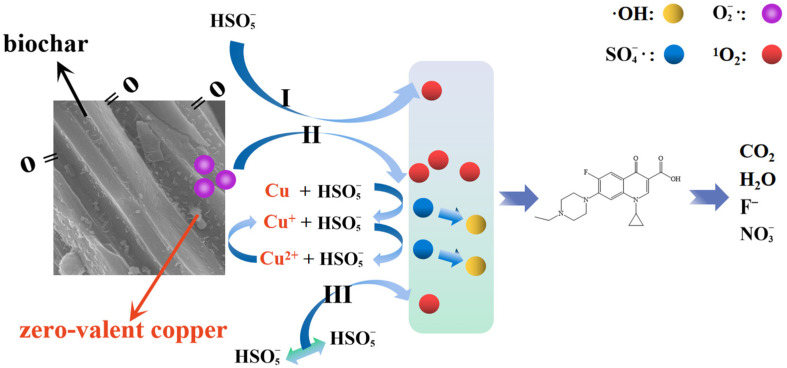
Schematic illustration of degradation mechanism in the CuC/PMS system.

**Figure 9 nanomaterials-12-02842-f009:**
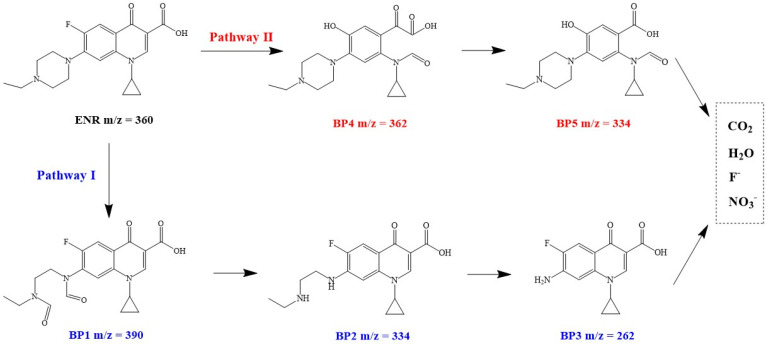
Possible intermediates and proposed transformation degradation pathways of ENR in CuC/PMS system.

**Table 1 nanomaterials-12-02842-t001:** Elemental composition of Fresh and used CuC.

	EDS				
	C	O	Cu	K	Cl
Fresh CuC	83.6	6.9	2.4	3.1	4.1
Used CuC	88.6	7.8	0.5	0.6	2.5

## Data Availability

Not applicable.

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
