# Peer review of "Efficient Activation of Peroxymonosulfate by Biochar-Loaded Zero-Valent Copper for Enrofloxacin Degradation: Singlet Oxygen-Dominated Oxidation Process"

_nanomaterials, 2022, doi:10.3390/nano12162842_

Round 1

Reviewer 1 Report

This paper reported the degradation of enrofloxacin using PMS activated by biochar-loaded zero-valent copper. Complete degradation of the pollutants was reported with the proposed method. They could also identify the various ROS such as SO4●-, ●OH, O2●-, and 1O2 in the PMS-activated system. The results are presented well and therefore it can be accepted after a major revision.

Abstract:

1.   The important observations/results are not given in the abstract. It seems to be just what they did in this study

2.   The symbols for radicals are not presented well.

Introduction

3.   There are many related studies reported for the removal of a variety of contaminants. The novelty needs to be highlighted in the introduction.

4.   The in situ generated reactive nitrogen species is a major issue for N2-containing pollutants. It should be addressed in the introduction, REF: https://doi.org/10.1016/j.cej.2021.133002

5.   For the thermal activation of PS please cite, https://doi.org/10.1016/j.coche.2022.100839

6.   In the section describing the activation by ZVI, please refer https://doi.org/10.1016/j.cej.2017.01.031

7.   Line 67 – 68, please explain shortly the reasons for the highest PMS activation ability of Cu compared to other transition metals, especially Fe

Materials and Methods

8.   Line no. 104, is it potassium persulfate or PMS ?

9.   Section 2.3, Explain the procedure for the scavenging experiment. What is the concentration of scavenger (or pollutant to scavenger ratio)

10.           Section 2.4, Please cite a reference for the spectrophometric determination of PMS

11.           Section 2.6, what is the detection wavelength for HPLC ?

12.           Please give the full name for EPR

Results and discussions

13.  Figure 1: It is better to explain in the in the same order given in the text. For eg: Fig 1 a SEM, so that the readers can easily understand

14.  Figure f is cited in the text, but, it is not given in the text

15.  Section 3.2, in the introduction it is mentioned that PMS is more active than PDS. Have they compared in the present study?

16.  In addition, I proposed to do some additional tests such as Cu+, and Cu2+ in the PMS activation for the degradation

17.  This section seems to be the presentation of some results, it will be nice if they could provide an explanation for the results.

18.  Section 3.3.1: This section initially gives an idea that major species are SO4- and OH radical. Line 231-232 is bit confusing. Because in the scavenging study, it is clear that major species in O12 and O2●-. Please explain it clearly.

19.  Please provide an equation for the generation of reactive species in this section (OH and SO4- radicals)

20.  Section 3.3.2: What is the Pka of ENR. Is there any effect on the structural changes of pollutants on the degradation efficiency?

21.  Section 3.3.3: Line 269, is there any reference for this statement?

22.  The bicarbonate ions addition increases the pH of the solution? Have they monitored any effect in the present study?

23.  Section 3.4.1, explain the scavenger to pollutant ratio in the caption of figure 6

24.  Line 306: it will be nice if there is an equation for the singlet oxygen generation from super oxide radical

25.  Line 311: Why PMS is consumed in the presence of pBQ and LHD than the control experiment?

26.  What is difference between the green line in the EPR (Fig 6e and f)? is there any change in the experimental condition?

27.  The symbols in the Figure 6 is confusing

28.  Section 3.4.2. Have they checked the activation of PMS by used materials?

29.  Have they noticed the leaching of Cu from the material during the degradation?

30.  It will be more informative and stronger the section 3.4.2 moves to section 3.3. In this case, the story will be clear

31.  Section 3.5, line 405, have they monitored quantitatively the F- and NO3- formation during the oxidation process?

32.  What about TOC reduction?

33.  It will be nice if the author could a TOC measurement in pure water (PMS activation in pure water).

34.  In the conclusion, it is mentioned “economical”…… have they done any such estimation?

Author Response

Journal: Nanomaterials

Manuscript ID: nanomaterials-1842432

Title: "Efficient activation of peroxymonosulfate by biochar-loaded zero-valent copper for enrofloxacin degradation: Singlet oxy-gen-dominated oxidation process"

Author(s): Jiang Zhao, Tianyin Chen, Cheng Hou, Baorong Huang, Jiawen Du, Nengqian Liu, Xuefei Zhou and Yalei Zhang

Note:

  • Our responses to the reviewers’ comments are shown in color blue.
  • All actions of editing taken are shown in color red.
  • The line and page numbers referred correspond to those in the revised manuscript.

Reviewer(s)' Comments to Author:

Reviewer #1:

This paper reported the degradation of enrofloxacin using PMS activated by biochar-loaded zero-valent copper. Complete degradation of the pollutants was reported with the proposed method. They could also identify the various ROS such as SO4●-, ●OH, O2●-, and 1O2 in the PMS-activated system. The results are presented well and therefore it can be accepted after a major revision.

Abstract:

  1. The important observations/results are not given in the abstract. It seems to be just what they did in this study

Response: Thanks a lot for your constructive comments. We have improved the abstract of this paper.

Action: Please review the abstract.

  1. The symbols for radicals are not presented well.

Response: We are sorry that the symbols for radicals was not accurate in the original manuscript. We have revised them again.

Action: Please check abstract, section 3.4 and conclusion.

Introduction

  1. There are many related studies reported for the removal of a variety of contaminants. The novelty needs to be highlighted in the introduction.

Response: Thanks a lot for your constructive comments. There have few studies have reported the activation of PMS by BC-loaded Cu0 (CuC), and the mechanism by which the CuC/PMS system degrades pollutants still needs further investigation. This work confirmed that the reactive oxygen species (ROS) involved in ENR degradation were ·OH, SO- 4·, 1O2, and O- 2·. Among them, 1O2 played a major role in degradation, whereas O- 2· played a key role in the indirect generation of 1O2. On the one hand, CuC adsorbed and activated PMS to generate ·OH, SO- 4· and O- 2·. O- 2· was unstable and reacted rapidly with H2O and ·OH to generate large amounts of 1O2. On the other hand, both the self-decomposition of PMS and direct activation of PMS by C=O on biochar also generated 1O2.

Action: Please review the introduction and abstract.

  1. The in situgenerated reactive nitrogen species is a major issue for N2-containing pollutants. It should be addressed in the introduction, REF: https://doi.org/10.1016/j.cej.2021.133002

Response: Thanks a lot for your constructive comments. We haven't considered that before. You give us a creative direction. We will then conduct further research on the presence of reactive nitrogen species.

Action: Section 3.5: the following statement has been added.

“However, how to control the ecological toxicity of the degradation process of N-containing pollutants still needs more attention and in-depth research [1].”

  1. For the thermal activation of PS please cite, https://doi.org/10.1016/j.coche.2022.100839.

Response: Thanks a lot for your constructive comment.

Action: Line 59: add the citation [2].

  1. In the section describing the activation by ZVI, please refer https://doi.org/10.1016/j.cej.2017.01.031

Response: Thanks a lot for your constructive comment. In addition, the slow release of ROS catalyzed by Fe0 mentioned in this study fits our thinking.

Action: Line 66: add the citation [3].

  1. Line 67 – 68, please explain shortly the reasons for the highest PMS activation ability of Cu compared to other transition metals, especially Fe.

Response: Thanks a lot for your constructive comments. The aggregation of powders on the surface of the magnetic stirrer resulting from zero-valent iron’s magnetic susceptibility may have caused the relatively low removal rates. Additionally, the magnetic susceptibility may affect the application of catalyst in practice, so we think Cu may be better.

Action: Line 68-72, “It has been shown that zero-valent Cu (Cu0) can directly activate PMS to generate SO- 4· [4], and among the different zero-valent metal activation PMS systems, Cu has the highest efficiency in activating PMS [5]. Therefore, the use of Cu0 activating PMS system is worthy of our further study.”has been revised to“It has been shown that zero-valent Cu (Cu0) can directly activate PMS to generate SO- 4· [4], and in jar experiments, zero-valent Cu even has the higher efficiency in activating PMS than zero-valent iron because of zero-valent iron’s magnetic susceptibility [5]. Therefore, the use of Cu0 activating PMS system is worthy of our further study and application.”

Materials and Methods

  1. Line no. 104, is it potassium persulfate or PMS ?

Response: We are sorry that this phrase was not accurate in the original manuscript.

Action: Line no.107, “potassium persulfate” has been revised to “peroxymonosulfate”.

  1. Section 2.3, explain the procedure for the scavenging experiment. What is the concentration of scavenger (or pollutant to scavenger ratio).

Response: Thanks a lot for your constructive comments. The concentration of scavenger is critical in quenching experiments, which determines whether quenching is rapid and thorough. We did a series of gradient experiments for the four scavenger, and selected the concentrations with the most significant reaction inhibition effect, tert-butanol (TBA, 4 mmol/L), methanol (MeOH, 4 mmol/L), L-histidine (LHD,10 mmol/L), and p-benzoquinone (PBQ, 5mmol/L), respectively.

Action:

  • Line 128, the following statement has been added.

“For the quenching experiments, the scavengers were also added at this time.”.

(2) Line 346-347, “including tert-butanol (TBA), methanol (MeOH), L-histidine (LHD), and p-benzoquinone (PBQ)” has been revised to “including tert-butanol (TBA, 4 mmol/L), methanol (MeOH, 4 mmol/L), L-histidine (LHD, 10 mmol/L), and p-benzoquinone (PBQ,5 mmol/L)”.

  1. Section 2.4, Please cite a reference for the spectrophometric determination of PMS.

Response: We are sorry that the related article was not explicitly cited in the original manuscrip.

Action: Section 2.4,“As per previous studies [38], the PMS concentration in the solution was determined by the yellow iodine color formed from KI. NaHCO3 (1.5 g) and KI (30 g) were added to 300 mL of deionized water and mixed thoroughly as a background solution. A portion of the reaction solution (0.5 mL) was extracted, and then added in a 10-mL colorimetric tube. The reaction solution was fixed to a scale using a background solution and shaken for 15 min. The final analysis was performed at a wavelength of 352 nm by using a UV spectrophotometer.” has been revised to “The PMS concentration in the solution was determined by the yellow iodine color formed from KI. NaHCO3 (1.5 g) and KI (30 g) were added to 300 mL of deionized water and mixed thoroughly as a background solution. A portion of the reaction solution (0.5 mL) was extracted, and then added in a 10-mL colorimetric tube. The reaction solution was fixed to a scale using a background solution and shaken for 15 min. The final analysis was performed at a wavelength of 352 nm by using a UV spectrophotometer [38].”

  1. Section 2.6, what is the detection wavelength for HPLC ?

Response: Thanks a lot for your constructive comments.

Action:

  • Line 159, the following statement has been added.

“(detailed information provided in Table S1)”.

  • Supporting Information, add Table S1.

Table S1. Details of the eluents and detection wavelengths of HPLC

Compound

Eluents

Wavelengths (nm)

Enrofloxacin

acetonitrile: (0.35% phosphoric acid + triethylamine, pH = 3.0) = 17: 83

277

  1. Please give the full name for EPR

Response: Thanks a lot for your constructive comments.

Action: Line 162, the following statement has been added.

“Electron paramagnetic resonance”

Results and discussions

  1. Figure 1: It is better to explain in the in the same order given in the text. For eg: Fig 1 a SEM, so that the readers can easily understand.

Response: Thanks a lot for your constructive comments.

Action: see Section 3.1 in manuscript.

  1. Figure f is cited in the text, but, it is not given in the text

Response: We are sorry that the citation was not correct in the original manuscrip.

Action: see Section 3.1 in manuscript.

  1. Section 3.2, in the introduction it is mentioned that PMS is more active than PDS. Have they compared in the present study?

Response: Thanks a lot for your constructive comments. Compared with the differences between PDS and PMS, our study focused on the activation effect and activation mechanisms of biochar-loaded zero-valent copper. Regarding the stronger reactivity of PMS than that of PDS, we refer to zhou’s study[5]。

Action: No further actions.

  1. In addition, I proposed to do some additional tests such as Cu+, and Cu2+ in the PMS activation for the degradation.

Response: Thanks a lot for your constructive comments. According to our analysis, the valence of Cu0 increases during activation (Cu0 → Cu+ → Cu2+), so that it can gradually activate PMS to produce ROS. Cu+ and Cu2+ are actually the intermediates of the activation process, so we did not study them alone. However, when laboratory conditions permit, we will definitely conduct the experiments. The results may be interesting.

Action: No further actions.

  1. This section seems to be the presentation of some results, it will be nice if they could provide an explanation for the results.

Response: Thanks a lot for your constructive comments. In section3.2, our main aim is to demonstrate the superiority of the CuC / PMS system over the other one-component systems. What’s more, we also propose the following speculation through the comparison: “ the loading of Cu ions (such as Cu+ and Cu2+) can form a complexation reaction between transition metal ions and organic matter, which increases the chemisorption of ENR [6]”、“BC is often used as an effective catalyst for PMS-based AOPs [7], and it can directly activate PMS through C=O bonding to produce 1O2, which leads to the oxidative decomposition of organic pollutants [4].” and “The degradation efficiency of ENR in the (Cu + BC)/PMS system (using copper powder) reached 89%, indicating that Cu0 was the main species for catalytic activation of PMS in this system.”. And the specific analysis is developed in Section 3.4.

Action: No further actions.

  1. Section 3.3.1: This section initially gives an idea that major species are SO4- and OH radical. Line 231-232 is bit confusing. Because in the scavenging study, it is clear that major species in O12 and O2●-. Please explain it clearly.

Response: Thanks a lot for your constructive comments. We are sorry that there was ambiguity in the original manuscript. Self-quenching of the excess PMS is only involved SO- 4· and ·OH. When the CuC dosage was fixed, with the excessive increase in PMS dosages, the degradation efficiency decreased only 3.9 % in 30 min, which also indicated that SO- 4· and ·OH did not play a major role in this system.

Action: Section 3.3.1, “In the first 30 min, the degradation effect at 3 mM was better than that at 4 mM. Raising the PMS dosage within a certain range could lead to an increase in ROS concentration, which would promote the degradation of ENR. However, excessive PMS transiently generates high concentrations of ROS, resulting in the self-quenching reactions shown in Eqs. (1) and (2) [8, 9] to form SO5-· and HSO- 4, respectively, with lower reactivities.” has been revised to “In the first 30 min, the degradation effect at 3 mM was slightly better than that at 4 mM. Raising the PMS dosage within a certain range could lead to an increase in ROS concentration, which would promote the degradation of ENR. However, excessive PMS transiently generates certain ROS (SO- 4· and ·OH) with high concentrations, resulting in the self-quenching reactions shown in Eqs. (1) and (2) [8, 9] to form SO5-· and HSO- 4, respectively, with lower reactivities.”

  1. Please provide an equation for the generation of reactive species in this section (OH and SO4- radicals)

Response: Thanks a lot for your constructive comments.The related equation has been shown in Section 3.4.2.. But Moving them to the place you indicate will be indeed more intuitive.

Action: See the equation (1)~(2) in Section 3.3.1. in manuscript.

  1. Section 3.3.2: What is the Pka of ENR. Is there any effect on the structural changes of pollutants on the degradation efficiency?

Response: ENR has two different acid dissociation constant (pKa) values, exists in cationic form at pH < 6.20, zweitter anion species at 6.20 < pH< 7.80 and anion form at pH > 7.80 [10]. When PH> 9, ENR and CuC have an electrostatic repulsion. However, ENR was degraded by the oxidation of ROS in CuC/PMS system , and the degradation of ENR was not a non-radical pathway in which CuC was a electron carrier between PMS and ENR. Therefore, CuC adsorbing PMS and activating PMS to produce ROS were critical steps to affect the degradation rate, while the relationship of ENR and CuC has relatively little effect on the degradation efficiency.

Action: No further actions.

  1. Section 3.3.3: Line 269, is there any reference for this statement?

Response: Thanks a lot for your constructive comments. We have added related references.

Action: We cite two more articles to explain it, including [11]、[12].

  1. The bicarbonate ions addition increases the pH of the solution? Have they monitored any effect in the present study?

Response: Thanks a lot for your constructive comments. The bicarbonate ions addition increases the pH of the solution. But based on the experimental data , the effect of pH was smaller than the reaction between HCO- 3 and ROS.

The concentration of bicarbonate ions (mmol/L)

pH

0

3.12

1

3.84

5

6.34

30

8.22

150

10.25

Action: No further actions.

  1. Section 3.4.1, explain the scavenger to pollutant ratio in the caption of figure 6

Response: Thanks a lot for your constructive comments. Combined with question 9, we have modify the caption.

Action: Line 348: “Figure 6. Effect of ROS scavengers on the removal of ENR by CuC/PMS at (a) pH = 3, (b) pH = 7, and (c) pH = 9. Experimental conditions: [CuC]0 = 0.2 g/L, [ENR]0 = 10 mg/L, [PMS]0 = 2 mmol/L, T = 25 ℃. (d) Decomposition of PMS upon the removal of ENR by CuC/PMS. Experimental conditions: [CuC]0 = 0.2 g/L, [ENR]0 = 10 mg/L, [PMS]0 = 2 mmol/L, initial pH =3.10, T = 25 ℃. EPR spectra of (e) CuC/PMS and (f) PMS with the addition of DMPO and TEMP. Experimental conditions: [CuC]0 = 0.2 g/L, [ENR]0 = 10 mg/L, [PMS]0 = 2 mmol/L, initial pH =3.10, T = 25 ℃.” has been revised to “Figure 6. Effect of ROS scavengers (4 mmol/L TBA, 4 mmol/L MeOH, 10 mmol/L LHD and 5 mmol/L PBQ) on the removal of ENR by CuC/PMS at (a) pH = 3, (b) pH = 7, and (c) pH = 9. Experimental conditions: [CuC]0 = 0.2 g/L, [ENR]0 = 10 mg/L, [PMS]0 = 2 mmol/L, T = 25 ℃. (d) Decomposition of PMS upon the removal of ENR by CuC/PMS. Experimental conditions: [CuC]0 = 0.2 g/L, [ENR]0 = 10 mg/L, [PMS]0 = 2 mmol/L, initial pH =3.10, T = 25 ℃. EPR spectra of (e) CuC/PMS and (f) PMS with the addition of DMPO and TEMP. Experimental conditions: [CuC]0 = 0.2 g/L, [ENR]0 = 10 mg/L, [PMS]0 = 2 mmol/L, initial pH =3.10, T = 25 ℃.”

  1. Line 306: it will be nice if there is an equation for the singlet oxygen generation from super oxide radical.

Response: Thanks a lot for your constructive comments. The related equation has been shown in Section 3.4.2.. But Moving them to the place you indicate will be indeed more intuitive.

Action: Please review the equation (6) in Line 344 in manuscript.

  1. Line 311: Why PMS is consumed in the presence of pBQ and LHD than the control experiment?

Response: Thanks a lot for your constructive comments. According to Figure 6d, PMS was excessive in this experiment. In the reaction that PMS generates ROS, the scavenger can quickly react with ROS, resulting in further consumption of more reactants, PMS. Therefore, PMS is consumed in the presence of pBQ and LHD than the control experiment.

For TBA, it quenched ·OH, making equation (2) react in the positive direction, which produce a large amount of SO42-. A large amount of SO42- inhibit the reaction of the main equation (3). So PMS consumption is less than the control experiment.

(1)

(2)

(3)

Action: No further actions

  1. What is difference between the green line in the EPR (Fig 6e and f)? is there any change in the experimental condition?

Response: Thanks a lot for your constructive comments. Both green lines indicate the EPR spectra using spin-trapping for O- 2·.Fig 6e was under the system of Cu/C + PMS, which can observe the typical peaks of O- 2·.While Fig 6f was under the system of PMS alone, which can not observe the peaks of  O- 2.

Action: No further actions.

  1. The symbols in the Figure 6 is confusing.

Response: We are sorry that the symbols was not clearly in the original manuscrip.

Action: We have revised it in figure 6.

  1. Section 3.4.2. Have they checked the activation of PMS by used materials?

Response: Thanks a lot for your constructive comments. We were planning to check the activation of PMS in liquid and soil medium by used materials. Firstly, we will simulate the repeating utilization factor of Cu/C. Secondly, it will be tested to investigate degradation rate of Cu/C- PMS for ENR or other CECs in water and groundwater. However, we haven’t the laboratory and conditions to achieve any experiment affected by Omicron at Shanghai.

Action: No further actions.

  1. Have they noticed the leaching of Cu from the material during the degradation?

Response: Thanks a lot for your constructive comments. We have tested the solution (filtrated by 0.45 μm filter membrane) by ICP-MS after 90 min of reaction. The results showed that the content of Cu in the solution was 6.508 μg/L, which was far less than the theoretical content of copper in the prepared catalyst (49.16 mg/L).

Action: No further actions

  1. It will be more informative and stronger the section 3.4.2 moves to section 3.3. In this case, the story will be clear.

Response: Thanks a lot for your constructive comments.We have moved the equations in Section 3.4.2 to the corresponding position in Section 3.3, following your recommendations in question 21 and 24. And then we keep 3.4.2 as a summary.

Action: We have revised that. Please review question 21 and 24 in this document.

  1. Section 3.5, line 405, have they monitored quantitatively the F-and NO3- formation during the oxidation process?

Response: Thanks a lot for your constructive comments. We were planning to check the activation of PMS in liquid and soil medium by used materials. Firstly, we will simulate the repeating utilization factor of Cu/C. Secondly, it will be tested to investigate degradation rate of Cu/C- PMS for ENR or other CECs in water and groundwater. However, we haven’t the laboratory and conditions to achieve any experiment affected by Omicron at Shanghai.

Action: No further actions.

  1. 32. What about TOC reduction?

Response: Thanks a lot for your constructive comments. We were planning to do the TOC measurement. However, we haven’t the laboratory and conditions to achieve any experiment affected by Omicron at Shanghai. Our laboratory will reopen at 07/10/2022. And we will add this results in the next paper about Cu/C- PMS for ENR in groundwater used by column experiment.

Action: No further actions.

  1. It will be nice if the author could a TOC measurement in pure water (PMS activation in pure water).

Response: Thanks a lot for your constructive comments. We are planning do the TOC measurement in pure water (PMS activation in pure water). However, we haven’t the laboratory and conditions to achieve any experiment affected by Omicron at Shanghai. Our laboratory will reopen at 07/10/2022. And we will add this results in the next paper about Cu/C- PMS for ENR in groundwater used by column experiment.

Action: No further actions.

  1. In the conclusion, it is mentioned “economical”…… have they done any such estimation?

Response: Thanks a lot for your constructive comments. In this work, a simple pyrolysis method was designed to prepare a high-performance Cu/C material. From that perspective this Cu/C material is“economical”. In next work, we are planning to estimate the “economical” with column experiment.

Action: No further actions.

reference

  1. M.P. Rayaroth, C.T. Aravindakumar, N.S. Shah, et al., Advanced oxidation processes (AOPs) based wastewater treatment - unexpected nitration side reactions - a serious environmental issue: A review, Chem. Eng. J., 2022, 430, 133002.
  2. S. Sonawane, M.P. Rayaroth, V.K. Landge, et al., Thermally activated persulfate-based Advanced Oxidation Processes — recent progress and challenges in mineralization of persistent organic chemicals: a review, Curr. Opin. Chem. Eng., 2022, 37, 100839.
  3. M.P. Rayaroth, C.-S. Lee, U.K. Aravind, et al., Oxidative degradation of benzoic acid using Fe0- and sulfidized Fe0-activated persulfate: A comparative study, Chem. Eng. J., 2017, 315, 426-436.
  4. J.T. Shi, B.R. Dai, X.Y. Fang, et al., Waste preserved wood derived biochar catalyst for promoted peroxymonosulfate activation towards bisphenol A degradation with low metal ion release: The insight into the mechanisms, Sci. Total Environ., 2022, 813.
  5. P. Zhou, J. Zhang, Y. Zhang, et al., Degradation of 2,4-dichlorophenol by activating persulfate and peroxomonosulfate using micron or nanoscale zero-valent copper, J. Hazard. Mater., 2018, 344, 1209-1219.
  6. M. Graouer-Bacart, S. Sayen, E. Guillon, Macroscopic and molecular approaches of enrofloxacin retention in soils in presence of Cu(II), J. Colloid Interface Sci., 2013, 408, 191-199.
  7. W. Zhang, L.G. Yan, Q.D. Wang, et al., Ball milling boosted the activation of peroxymonosulfate by biochar for tetracycline removal, J. Environ. Chem. Eng., 2021, 9.
  8. C. Brandt, R. van Eldik, Transition metal-catalyzed oxidation of Sulfur(IV) oxides. atmospheric-relevant processes and mechanisms, Chem. Rev., 1995, 95, 119-190.
  9. J. Zou, J. Ma, L. Chen, et al., Rapid acceleration of ferrous iron/peroxymonosulfate oxidation of organic pollutants by promoting Fe(III)/Fe(II) cycle with hydroxylamine, Environ. Sci. Technol., 2013, 47, 11685-11691.
  10. T.S. Anirudhan, F. Shainy, J. Christa, Synthesis and characterization of polyacrylic acid- grafted-carboxylic graphene/titanium nanotube composite for the effective removal of enrofloxacin from aqueous solutions: Adsorption and photocatalytic degradation studies, J. Hazard. Mater., 2017, 324, 117-130.
  11. Z. Chen, S. Bi, G. Zhao, et al., Enhanced degradation of triclosan by cobalt manganese spinel-type oxide activated peroxymonosulfate oxidation process via sulfate radicals and singlet oxygen: Mechanisms and intermediates identification, Sci. Total Environ., 2020, 711, 134715.
  12. S. Wang, L. Xu, J. Wang, Nitrogen-doped graphene as peroxymonosulfate activator and electron transfer mediator for the enhanced degradation of sulfamethoxazole, Chem. Eng. J., 2019, 375, 122041.

Reviewer 2 Report

The manuscript is interesting and it requires some moderate revisions:

1)English grammar should be double-checked;

2)in general, the results discussion must be enlarged and more comparisons with literature data should be reported. The authors should go more in depth with the interpretation of the results;

3)the advancement of knowledge reached by the findings of the study must be underlined in the conclusion section;

4)the unit of measure of each parameter should follow the parameter itself when it appears in the text;

5)the practical implication of this study are not well covered by the authors;

6)the authorsshould report some data about the reusability of the synthetized material;

7)what about kinetic modelling?

Author Response

Journal: Nanomaterials

Manuscript ID: nanomaterials-1842432

Title: "Efficient activation of peroxymonosulfate by biochar-loaded zero-valent copper for enrofloxacin degradation: Singlet oxy-gen-dominated oxidation process"

Author(s): Jiang Zhao, Tianyin Chen, Cheng Hou, Baorong Huang, Jiawen Du, Nengqian Liu, Xuefei Zhou and Yalei Zhang

Note:

  • Our responses to the reviewers’ comments are shown in color blue.
  • All actions of editing taken are shown in color red.
  • The line and page numbers referred correspond to those in the revised manuscript.

Reviewer(s)' Comments to Author:

Reviewer #2:

  • English grammar should be double-checked;

Response: Thanks a lot for your constructive comment. We have double-checked the English grammar of the manuscript.

Action: The whole manuscript has been revised.

  • in general, the results discussion must be enlarged and more comparisons with literature data should be reported. The authors should go more in depth with the interpretation of the results;

Response: We greatly appreciate your constructive suggestion. In this work, we summarized some related work on the separate use of zero-valent copper/biochar to activate PMS in pollutants treatment. However, there have few of researches have been conducted to investigate the use of biochar-loaded zero-valent copper to activate PMS in pollutants removal. But, we are planning to do more explored about this system and go more in depth with the interpretation of the results.

Action: We will go more in depth with in next work.

  • the advancement of knowledge reached by the findings of the study must be underlined in the conclusion section;

Response: Thanks a lot for your constructive comment. We have improved the conclusion section of the manuscript.

Action: Please review the conclusion section.

4)the unit of measure of each parameter should follow the parameter itself when it appears in the text;

Response: Thanks a lot for your constructive comment. We have improved the manuscript.

Action: Please review the conclusion manuscript.

5)the practical implication of this study are not well covered by the authors;

Response: Thanks a lot for your constructive comments. In this work, a simple pyrolysis method was designed to prepare a high-performance Cu/C material. Researchers investigated the effectiveness and mechanism of Cu/C activation of PMS to degrade ENR. We were planning to do a column experiment and investigate practical implication of this study. However, we haven’t the laboratory and conditions to achieve any experiment affected by Omicron at Shanghai until our laboratory reopening at 07/10/2022.

Action: We are planning to do this work in next paper.

6)the authors should report some data about the reusability of the synthetized material;

Response: We greatly appreciate your constructive suggestion. We had tested the solution (filtrated by 0.45 μm filter membrane) by ICP-MS after 90 min of reaction. The results showed that the content of Cu in the solution was 6.508 μg/L, which was far less than the theoretical content of copper in the prepared catalyst (49.16 mg/L). Furthermore, we were planning to check the reusability of the synthetized material in liquid and soil medium. Firstly, we will simulate the repeating utilization factor of Cu/C. Secondly, it will be tested to investigate degradation rate of Cu/C- PMS for ENR or other CECs in water and groundwater. But, we haven’t the laboratory and conditions to achieve any experiment affected by Omicron at Shanghai before 07/10/2022.

Action: No further actions.

7)what about kinetic modelling?

Response: Thanks a lot for your constructive comment. We have revised and added kinetic modelling.

Action: In Section 3.2, fig.2 have been modified according to kinetic modelling.

Round 2

Reviewer 1 Report

The authors have adressed most of my suggestions in revising their manuscript.  However, the experiments for the further clarification of the results are not done. Other sections and revisions are in the stage of acceptace.